# BIT-REGULARIZED OPTIMIZATION OF NEURAL NETS

## ABSTRACT

We present a novel optimization strategy for training neural networks which we call "BitNet". The parameters of neural networks are usually unconstrained and have a dynamic range dispersed over all real values. Our key idea is to limit the expressive power of the network by dynamically controlling the range and set of values that the parameters can take. We formulate this idea using a novel end-to-end approach that regularizes a typical classification loss function. Our regularizer is inspired by the Minimum Description Length (MDL) principle. For each layer of the network, our approach optimizes real-valued translation and scaling factors and integer-valued parameters (weights). We empirically compare BitNet to an equivalent unregularized model on the MNIST and CIFAR-10 datasets. We show that BitNet converges faster to a superior quality solution. Additionally, the resulting model has significant savings in memory due to the use of integer-valued parameters.

## 1 INTRODUCTION

In the past few years, the intriguing fact that the performance of deep neural networks is robust to compact representations of parameters (Merolla et al., 2016; Zhou et al., 2016a), activations (Lin et al., 2015; Hubara et al., 2016) and even gradients (Zhou et al., 2016b) has sparked research in both machine learning (Gupta et al., 2015; Courbariaux et al., 2015; Han et al., 2015a) and hardware (Judd et al., 2016; Venkatesh et al., 2016; Hashemi et al., 2016). With the goal of improving the efficiency of deep networks on resource constrained embedded devices, learning algorithms were proposed for networks that have binary weights (Courbariaux et al., 2015; Rastegari et al., 2016), ternary weights (Yin et al., 2016; Zhu et al., 2016), two-bit weights (Meng et al., 2017), and weights encoded using any fixed number of bits (Shin et al., 2017; Mellempudi et al., 2017), that we herein refer to as the precision. Most of prior work apply the same precision throughout the network.

Our approach considers heterogeneous precision and aims to find the optimal precision configuration. From a learning standpoint, we ask the natural question of the optimal precision for each layer of the network and give an algorithm to directly optimize the encoding of the weights using training data. Our approach constitutes a new way of regularizing the training of deep neural networks, using the number of unique values encoded by the parameters (i.e. two to the number of bits) directly as a regularizer for the classification loss. To the best of our knowledge, we are not aware of an equivalent matrix norm used as a regularizer in the deep learning literature, forming a direct relationship between the low-level bit precision and regularization schemes inspired by the Minimum Description Length (MDL) principle.

*Outline and contributions:* Sec. 3 provides a brief background on our network parameters quantization approach, followed by a description of our model, BitNet. Sec. 4 contains the main contribution of the paper viz. (1) Convex and differentiable relaxation of the classification loss function over the discrete valued parameters, (2) Regularization using the complexity of the encoding, and (3) End-to-end optimization of a per-layer precision along with the parameters using stochastic gradient descent. Sec. 5 demonstrates faster convergence of BitNet to a superior quality solution compared to an equivalent unregularized network using floating point parameters. Sec. 6 we summarize our approach and discuss future directions.

## 2 RELATIONSHIP TO PRIOR WORK

There is a rapidly growing set of neural architectures (He et al., 2016; Szegedy et al., 2016) and strategies for learning (Duchi et al., 2011; Kingma & Ba, 2014; Ioffe & Szegedy, 2015) that study efficient learning in deep networks. The complexity of encoding the network parameters has been explored in previous works on network compression (Han et al., 2015a; Choi et al., 2016; Agustsson et al., 2017). These typically assume a pre-trained network as input, and aim to reduce the degradation in performance due to compression, and in general are not able to show faster learning. Heuristic approaches have been proposed (Han et al., 2015a; Tu et al., 2016; Shin et al., 2017) that alternate clustering and fine tuning of parameters, thus fixing the number of bits via the number of cluster centers. For example, the work of (Wang & Liang, 2016) assigns bits to layers in a greedy fashion given a budget on the total number of bits. In contrast, we use an objective function that combines both steps and allows an end-to-end solution without directly specifying the number of bits. Furthermore, we are able to show improved convergence over non bit-regularized networks training from scratch.

Our approach is closely related to optimizing the rate-distortion objective (Agustsson et al., 2017). In contrast to the entropy measure in (Choi et al., 2016; Agustsson et al., 2017), our distortion measure is simply the number of bits used in the encoding. We argue that our method is a more direct measure of the encoding cost when a fixed-length code is employed instead of an optimal variable-length code as in (Han et al., 2015a; Choi et al., 2016). Our work generalizes the idea of weight sharing as in (Chen et al., 2015a;b), wherein randomly generated equality constraints force pairs of parameters to share identical values. In their work, this 'mask' is generated offline and applied to the pre-trained network, whereas we dynamically vary the number of constraints, and the constraints themselves. A probabilistic interpretation of weight sharing was studied in (Ullrich et al., 2017) penalizing the loss of the compressed network by the KL-divergence from a prior distribution over the weights. In contrast to (Choi et al., 2016; Lin et al., 2016; Agustsson et al., 2017; Ullrich et al., 2017), our objective function makes no assumptions on the probability distribution of the optimal parameters. Weight sharing as in our work as well as the aforementioned papers generalizes related work on network pruning (Wan et al., 2013; Collins & Kohli, 2014; Han et al., 2015b; Jin et al., 2016; Kiaee et al., 2016; Li et al., 2016) by regularizing training and compressing the network without reducing the total number of parameters.

Finally, related work focuses on low-rank approximation of the parameters (Rigamonti et al., 2013; Denton et al., 2014; Tai et al., 2015; Anwar et al., 2016). This approach is not able to approximate some simple filters, e.g., those based on Hadamard matrices and circulant matrices (Lin et al., 2016), whereas our approach can easily encode them because the number of unique values is small[1]. Empirically, for a given level of compression, the low-rank approximation is not able to show faster learning, and has been shown to have inferior classification performance compared to approaches based on weight sharing (Gong et al., 2014; Chen et al., 2015a;b). Finally, the approximation is not directly related to the number of bits required for storage of the network.

## 3 NETWORK PARAMETERS QUANTIZATION

Though our approach is generally applicable to any gradient-based parameter learning for machine learning such as classification, regression, transcription, or structured prediction. We restrict the scope of this paper to Convolutional Neural Networks (CNNs).

**Notation.** We use boldface uppercase symbols to denote tensors and lowercase to denote vectors. Let $\mathbf{W}$ be the set of all parameters $\{\mathbf{W}^{(1)}, \dots, \mathbf{W}^{(N)}\}$ of a CNN with $N$ layers, $\tilde{\mathbf{W}}$ be the quantized version using $\mathcal{B}$ bits, and $w$ is an element of $\mathbf{W}$. For the purpose of optimization, assume that $\mathcal{B}$ is a real value. The proposed update rule (Section 4) ensures that $\mathcal{B}$ is a whole number. Let $\mathbf{y}$ be the ground truth label for a mini-batch of examples corresponding to the input data $\mathbf{X}^{(1)}$. Let $\hat{\mathbf{y}}$ be the predicted label computed as the maximum value in $\mathbf{X}^{(N)}$.

**Quantization.** Our approach is to limit the number of unique values taken by the parameters $\mathbf{W}$. We use a linear transform that uniformly discretizes the range into fixed steps $\delta$. We then quantize the

---

[1]Hadamard matrices can be encoded with 1-bit, Circulant matrices bits equal to log of the number of columns.

weights as shown in (1).

$$\tilde{\mathbf{W}} = \alpha + \delta \times \text{round}\left(\frac{\mathbf{W} - \alpha}{\delta}\right),$$

$$\text{where,} \quad \alpha = \min_w(\mathbf{W}^{(n)}), \qquad \beta = \max_w(\mathbf{W}^{(n)}), \qquad \delta = f(\mathbf{W}^{(n)}, \mathcal{B}) = \frac{\beta - \alpha}{2^{\mathcal{B}}}. \tag{1}$$

and all operations are elementwise operations. For a given range $\alpha, \beta$ and bits $\mathcal{B}$, the quantized values are of the form $\alpha + z\delta$, $z = 0, 1, \ldots, 2^B$, a piecewise constant step-function over all values in $[\alpha, \beta]$, with discontinuities at multiples of $\alpha + z\delta/2$, $z = 1, 3, \ldots, 2^B - 1$, i.e. $\mathbf{W}$ is quantized as $\tilde{\mathbf{W}} = \alpha + \delta\mathbf{Z}$. However, the sum of squared quantization errors defined in (2) is a continuous and piecewise differentiable function:

$$\mathcal{Q}(\mathbf{W}^{(n)}, \mathcal{B}) = \frac{1}{2}||\tilde{\mathbf{W}}^{(n)} - \mathbf{W}^{(n)}||_2^2. \tag{2}$$

where the 2-norm is an entrywise norm on the vector $vec(\tilde{\mathbf{W}}^{(n)} - \mathbf{W}^{(n)})$. For a given range $\alpha, \beta$ and $\mathcal{B}$, the quantization errors form a contiguous sequence of parabolas when plotted against all values in $[\alpha, \beta]$, and its value is bounded in the range $[0, \delta^2/8]$. Furthermore, the gradient with respect to $\mathcal{B}$ is $\frac{\partial \mathcal{Q}(\mathbf{W}^{(n)}, \mathcal{B})}{\partial \mathcal{B}} = (\tilde{\mathbf{W}}^{(n)} - \mathbf{W}^{(n)})\frac{\partial \delta}{\partial \mathcal{B}}\mathbf{Z}$ with $\frac{\partial \delta}{\partial \mathcal{B}} = -\delta$. The gradient with respect to $w \in \mathbf{W}^{(n)}$ is $\frac{\partial \mathcal{Q}(\mathbf{W}^{(n)}, \mathcal{B})}{\partial w} = (\tilde{\mathbf{W}}^{(n)} - \mathbf{W}^{(n)})(0 - 1)$.

In the above equations, the "bins" of quantization are uniformly distributed between $\alpha$ and $\beta$. Previous work has studied alternatives to uniform binning e.g. using the density of the parameters (Han et al., 2015a), Fisher information (Tu et al., 2016) or K-means clustering (Han et al., 2015a; Choi et al., 2016; Agustsson et al., 2017) to derive suitable bins. The uniform placement of bins is asymptotically optimal for minimizing the mean square error (the error falls at a rate of $O(1/2^B)$) regardless of the distribution of the optimal parameters (Gish & Pierce, 1968). Uniform binning has also been shown to be empirically better in prior work (Han et al., 2015a). As we confirm in our experiments (Section 5), such statistics of the parameters cannot guide learning when training starts from randomly initialized parameters.

## 4 LEARNING

The goal of our approach is to learn the number of bits jointly with the parameters of the network via backpropagation. Given a batch of independent and identically distributed data-label pairs $(\mathbf{X}^{(1)}, \mathbf{y})$, the loss function defined in (3) captures the log-likelihood.

$$l(\mathbf{W}) = -\log P(\mathbf{y}|\mathbf{X}^{(1)}; \mathbf{W}), \tag{3}$$

where the log likelihood is approximated by the average likelihood over batches of data. We cannot simply plug in $\tilde{\mathbf{W}}$ in the loss function (3) because $\tilde{\mathbf{W}}$ is a discontinuous and non-differentiable mapping over the range of $\mathbf{W}$. Furthermore, $\tilde{\mathbf{W}}$ and thus the likelihood using $\tilde{\mathbf{W}}$ remains constant for small changes in $\mathbf{W}$, causing gradient descent to remain stuck. Our solution is to update the high precision parameters $\mathbf{W}$ with the constraint that the quantization error $\mathcal{Q}(\mathbf{W}, \mathcal{B})$ is small. The intuition is that when $\mathbf{W}$ and $\tilde{\mathbf{W}}$ are close, $l(\mathbf{W})$ can be used instead. We adopt layer-wise quantization to learn one $\tilde{\mathbf{W}}^{(n)}$ and $\mathcal{B}^{(n)}$ for each layer $n$ of the CNN. Our new loss function $l(\tilde{\mathbf{W}})$ defined in (4) as a function of $l(\mathbf{W})$ defined in (3) and $\mathcal{Q}(\mathbf{W}, \mathcal{B})$ defined in (2).

$$l(\tilde{\mathbf{W}}) \triangleq l(\mathbf{W}) + \lambda_1 \sum_{i=1}^N \mathcal{Q}(\mathbf{W}^{(i)}, \mathcal{B}^{(i)}) + \lambda_2 \sum_{i=1}^N 2^{\mathcal{B}^{(i)}}, \tag{4}$$

where, $\lambda_1$ and $\lambda_2$ are the hyperparameters used to adjust the trade-off between the two objectives. When $\lambda_1 = 0, \lambda_2 = 1$, the CNN uses 1-bit per layer due to the bit penalty. When $\lambda_1 = 1, \lambda_2 = 0$, the CNN uses 32 bits per layer in order to minimize the quantization error. The parameters $\lambda_1$ and $\lambda_2$ allow flexibility in specifying the cost of bits vs. its impact on quantization and classification errors.

During training, first we update $\mathbf{W}$ and $\mathcal{B}$ using (5),

$$
\begin{aligned}
\mathbf{W}^{(i)} &\leftarrow \mathbf{W}^{(i)} - \mu \cdot \frac{\partial l(\tilde{\mathbf{W}})}{\partial \mathbf{W}^{(i)}} = \mathbf{W}^{(i)} - \mu \cdot \frac{\partial l(\mathbf{W})}{\partial \mathbf{W}^{(i)}} - \mu\lambda_1 \cdot \frac{\partial \mathcal{Q}(\mathbf{W}^{(i)}, \mathcal{B}^{(i)})}{\partial \mathbf{W}^{(i)}}, \\
\mathcal{B}^{(i)} &\leftarrow \mathcal{B}^{(i)} - \text{sign}(\mu\frac{\partial l(\tilde{\mathbf{W}})}{\partial \mathcal{B}^{(i)}}) = \mathcal{B}^{(i)} - \text{sign}(\mu\lambda_1\frac{\partial \mathcal{Q}(\mathbf{W}^{(i)}, \mathcal{B}^{(i)})}{\partial \mathcal{B}^{(i)}} + \mu\lambda_2 2^{\mathcal{B}^{(i)}}).
\end{aligned}
\tag{5}
$$

where $\mu$ is the learning rate. Then, the updated value of $\mathbf{W}$ in (5) is projected to $\tilde{\mathbf{W}}$ using $\mathcal{B}$ as in (1). The $sign$ function returns the sign of the argument unless it is $\epsilon$-close to zero in which case it returns zero. This allows the number of bits to converge as the gradient and the learning rate goes to zero [2]. Once training is finished, we throw away the high precision parameters $\mathbf{W}$ and only store $\tilde{\mathbf{W}}$ in the form of $\alpha + \delta \mathbf{Z}$ for each layer. All parameters of the layer can be encoded as integers, corresponding to the index of the bin, significantly reducing the storage requirement.

Overall, the update rule (5) followed by quantization (1) implements a projected gradient descent algorithm. From the perspective of $\tilde{\mathbf{W}}$, this can be viewed as clipping the gradients to the nearest step size of $\delta$. A very small gradient does not change $\tilde{\mathbf{W}}$ but incurs a quantization error due to change in $\mathbf{W}$. In practice, the optimal parameters in deep networks are bimodal (Merolla et al., 2016; Han et al., 2015b). Our training approach encourages the parameters to form a multi-modal distribution with means as the bins of quantization.

Note that $l(\tilde{\mathbf{W}})$ is a convex and differentiable relaxation of the negative log likelihood with respect to the quantized parameters. It is clear that $l(\tilde{\mathbf{W}})$ is an upper bound on $l(\mathbf{W})$, and $l(\tilde{\mathbf{W}})$ is the Lagrangian corresponding to constraints of small quantization error using a small number of bits. The uniformly spaced quantization allows a closed form for the number of unique values taken by the parameters. To the best of our knowledge, there is no equivalent matrix norm that can be used for the purpose of regularization.

## 5 EXPERIMENTS

We evaluate our algorithm, 'BitNet', on two benchmarks used for image classification, namely MNIST and CIFAR-10. *MNIST*: The MNIST (LeCun et al., 1998) database of handwritten digits has a total of $70,000$ grayscale images of size $28 \times 28$. We used $50,000$ images for training and $10,000$ for testing. Each image consists of one digit among $0, 1, \ldots, 9$. *CIFAR-10*: The CIFAR-10 (Krizhevsky, 2009) dataset consists of $60,000$ color images of size $32 \times 32$ in 10 classes, with $6,000$ images per class corresponding to object prototypes such as 'cat', 'dog', 'bird', etc. We used $40,000$ images for training and $10,00$ images for testing. The training data was split into batches of $200$ images.

**Setup.** For BitNet, we evaluate the error on the test set using the quantized parameters $\tilde{\mathbf{W}}$. We also show the training error in terms of the log-likelihood using the non-quantized parameters $\mathbf{W}$. We used the automatic differentiation provided in Theano (Bergstra et al., 2010) for calculating gradients with respect to (4).

Since our motivation is to illustrate the superior anytime performance of bit regularized optimization, we use a simple neural architecture with a small learning rate, do not train for many epochs or till convergence, and do not compare to the state-of-the-art performance. We do not perform any preprocessing or data augmentation for a fair comparison.

We use an architecture based on the LeNet-5 architecture (LeCun et al., 1998) consisting of two convolutional layers each followed by a $2 \times 2$ pooling layer. We used 30 and 50 filters of size $5 \times 5$ for MNIST, and 256 and 512 filters of size $3 \times 3$ for CIFAR-10 respectively. The filtered images are fed into a dense layer of $500$ hidden units for MNIST ($1024$ units for CIFAR-10 respectively), followed by a softmax layer to output scores over 10 labels. All layers except the softmax layer use the hyper-tangent ($tanh$) activation function. We call this variant 'LeNet32'.

**Baselines.** In addition to LeNet32, we evaluate three baselines from the literature that use a globally fixed number of bits to represent the parameters of every layer. The baselines are (1) 'linear-$n$': quantize the parameters at the end of each epoch using 1 (2) 'kmeans-$n$': quantize the parameters at the end of each epoch using the K-Means algorithm with $2^n$ centers, and (3) 'bayes-$n$': use the bayesian compression algorithm of (Ullrich et al., 2017). The baseline 'kmeans-$n$' is the basic idea behind many previous approaches that compress a pre-trained neural model including (Han et al., 2015a; Choi et al., 2016). For 'bayes-$n$' we used the implementation provided by the authors but without pre-training.

---

[2]In our experiments we use $\epsilon = 10^{-9}$.

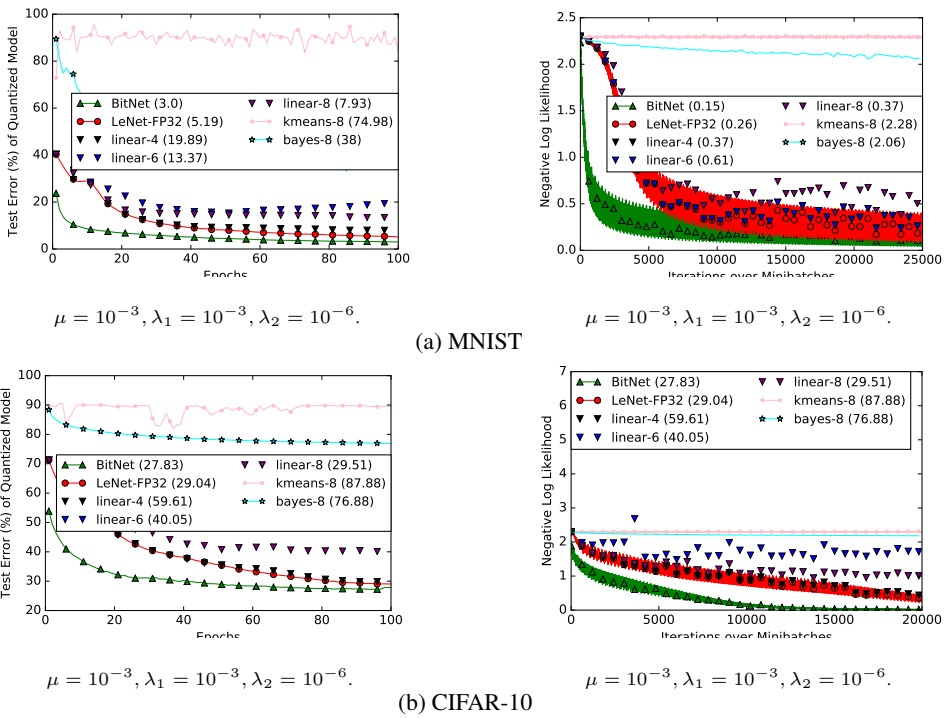

Figure 1: Performance of BitNet in comparison to LeNet32 and baselines on the MNIST and CIFAR-10 datasets. The left panel shows the test error % and the right panel shows the training error over training iterations. The learning rate $\mu$ is halved after every 200 iterations.

**Impact of bit regularization.** Figure 1 compares the performance of BitNet with our baselines. In the left panel, the test error is evaluated using quantized parameters. The test error of BitNet reduces more rapidly than LeNet32. The test error of BitNet after 100 epochs is 2% lower than LeNet32. For a given level of performance, BitNet takes roughly half as many iterations as LeNet32. As expected, the baselines 'kmeans8' and 'bayes8' do not perform well without pre-training. The test error of 'kmeans8' oscillates without making any progress, whereas 'bayes8' is able to make very slow progress using eight bits. In contrast to 'kmeans8', the test error of 'linear8' is very similar to 'LeNet32' following the observation by (Gupta et al., 2015) that eight bits are sufficient if uniform bins are used for quantization. The test error of 'BitNet' significantly outperforms that of 'linear8' as well as the other baselines. The right panel shows that the training error in terms of the log likelihood of the non-quantized parameters. The training error decreases at a faster rate for BitNet than LeNet32, showing that the lower testing error is not caused by quantization alone. This shows that the modified loss function of 4 has an impact on the gradients wrt the non-quantized parameters in order to minimize the combination of classification loss and quantization error. The regularization in BitNet leads to faster learning. In addition to the superior performance, BitNet uses an average of 6 bits per layer corresponding to a 5.33x compression over LeNet32. Figure 2 shows the change in the number of bits over training iterations. We see that the number of bits converge within the first five epochs. We observed that the gradient with respect to the bits quickly goes to zero.

**Sensitivity to Learning Rate.** We show that the property of faster learning in BitNet is indirectly related to the learning rate. For this experiment, we use a linear penalty for the number of bits instead of the exponential penalty (third term) in (4). In the left panel of Figure 3(a), we see that BitNet converges faster than LeNet-FP32 similar to using exponential bit penalty. However, as shown in the figure on the right panel of 3(a), the difference vanishes when the global learning rate is increased tenfold. This point is illustrated further in Figure 3(b) where different values of $\lambda_2$, the coefficient for the number of bits, shows a direct relationship with the rate of learning. Specifically, the right panel shows that a large value of the bit penalty $\lambda_2$ leads to instability and poor performance, whereas a smaller value leads to a smooth learning curve. However, increasing the learning rate globally as in LeNet-FP32 is not as robust as the adaptive rate taken by each parameter as in BitNet. Furthermore,

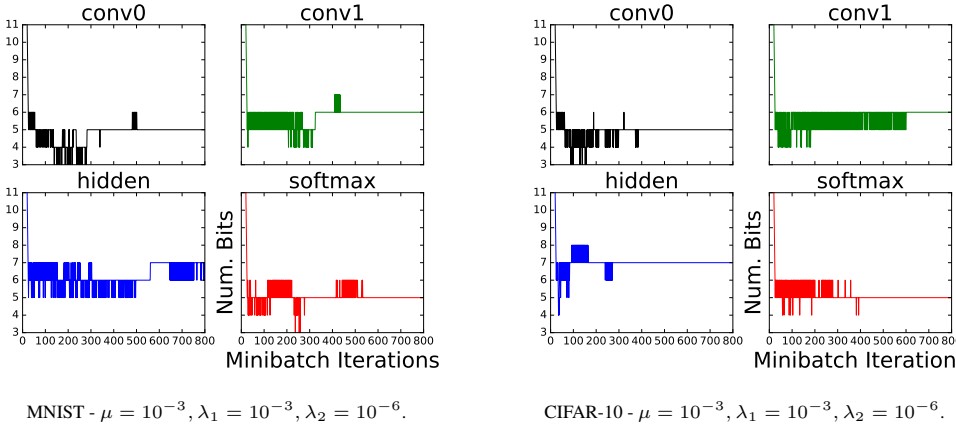

MNIST - $\mu = 10^{-3}, \lambda_1 = 10^{-3}, \lambda_2 = 10^{-6}$.

CIFAR-10 - $\mu = 10^{-3}, \lambda_1 = 10^{-3}, \lambda_2 = 10^{-6}$.

Figure 2: Number of bits learned by BitNet for representing the parameters of each layer of the CNN for MNIST and CIFAR-10.

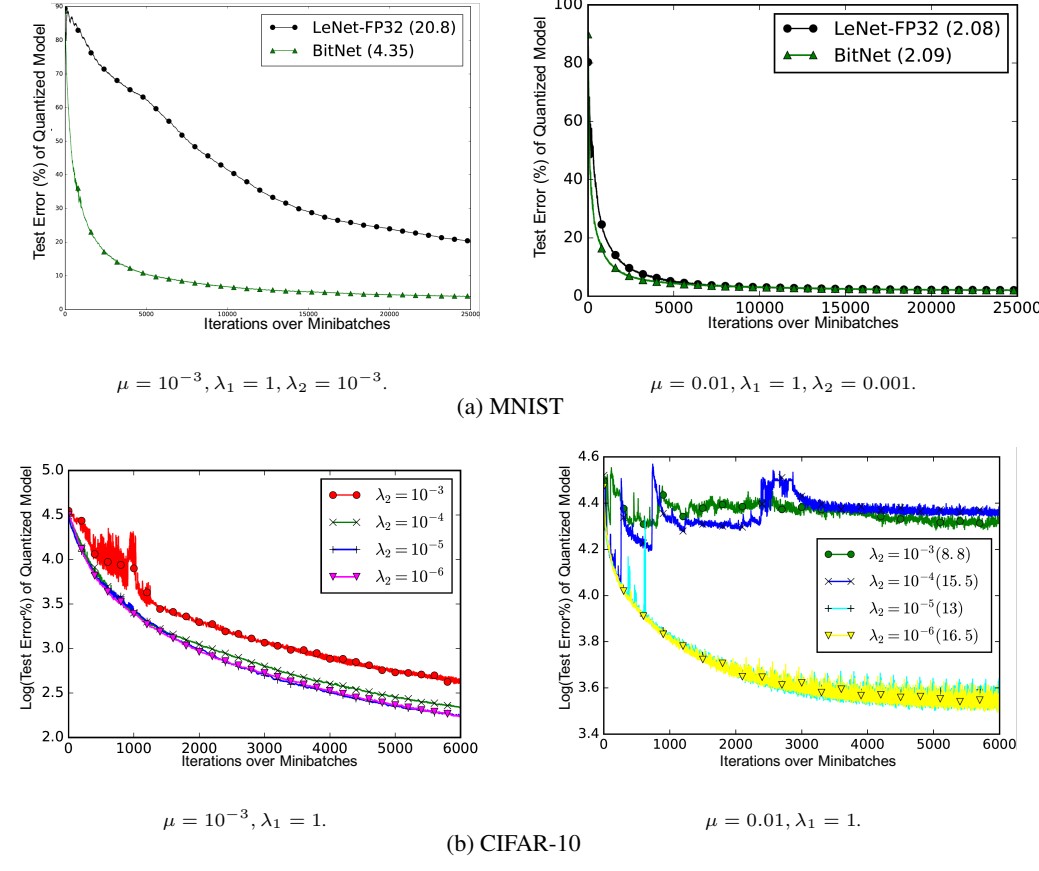

$\mu = 10^{-3}, \lambda_1 = 1, \lambda_2 = 10^{-3}$.

$\mu = 0.01, \lambda_1 = 1, \lambda_2 = 0.001$.

(a) MNIST

$\mu = 10^{-3}, \lambda_1 = 1$.

$\mu = 0.01, \lambda_1 = 1$.

(b) CIFAR-10

Figure 3: Sensitivity of BitNet to Learning Rate.

the learning oscillates especially if the global learning rate is further increased. This establishes an interesting connection between low precision training and momentum or other sophisticated gradient descent algorithms such as AdaGrad, which also address the issue of 'static' learning rate.

**Impact of Number of Layers.** In this experiment, we add more layers to the baseline CNN and show that bit regularization helps to train deep networks quickly without overfitting. We show a sample of a sequence of layers that can be added incrementally such that the performance improves.

Table 1: Performance of BitNet at the end of 30 epochs on MNIST and at the end of 100 epochs on CIFAR-10 with increasing complexity of the neural architecture. The first column (#) denotes the number of total layers. Compression is the ratio 32 to the average number of bits used by BitNet. The columns to the right of this ratio specifies the architecture and the number of bits in the final BitNet model. The column heading for a convolutional layer specifies the number of filters, the spatial domain and the pooling size. Here $\lambda_1 = 10^{-3}, \lambda_2 = 10^{-6}$.

**MNIST**

| # | Test Error % | Num. Params. | Compr. Ratio | 30 $1 \times 1$ $1 \times 1$ | 30 $3 \times 3$ $1 \times 1$ | 30 $3 \times 3$ $2 \times 2$ | 30 $5 \times 5$ $4 \times 4$ | 50 $5 \times 5$ $4 \times 4$ | Dense 500 Nodes | Dense 500 Nodes | Classify 10 Labels |
|---|---|---|---|---|---|---|---|---|---|---|---|
| 4 | 11.16 | 268K | 6.67 | | | | 5 | 6 | 7 | | 6 |
| 5 | 10.46 | 165K | 5.72 | | | 5 | 6 | 6 | 6 | | 5 |
| 6 | 9.12 | 173K | 5.65 | | 5 | 6 | 6 | 6 | 6 | | 5 |
| 7 | 8.35 | 181K | 5.75 | 5 | 5 | 6 | 6 | 6 | 6 | | 5 |
| 8 | 7.21 | 431K | 5.57 | 5 | 6 | 5 | 6 | 6 | 6 | 7 | 5 |

**CIFAR-10**

| # | Test Error % | Num. Params. | Compr. Ratio | 30 $1 \times 1$ $1 \times 1$ | 30 $3 \times 3$ $1 \times 1$ | 30 $3 \times 3$ $2 \times 2$ | 50 $3 \times 3$ $2 \times 2$ | 50 $3 \times 3$ $2 \times 2$ | Dense 500 Nodes | Dense 500 Nodes | Classify 10 Labels |
|---|---|---|---|---|---|---|---|---|---|---|---|
| 4 | 42.08 | 2.0M | 5.57 | | | 5 | 6 | | 7 | | 5 |
| 5 | 43.71 | 666K | 5.52 | | | 5 | 6 | 6 | 7 | | 5 |
| 6 | 43.32 | 949K | 5.49 | | 5 | 5 | 6 | 6 | 7 | | 6 |
| 7 | 42.83 | 957K | 5.74 | 4 | 5 | 6 | 6 | 6 | 7 | | 5 |
| 8 | 41.23 | 1.2M | 5.57 | 4 | 5 | 6 | 6 | 6 | 7 | 7 | 5 |

We selected these layers by hand using intuition and some experimentation. Table 1 shows the results for the MNIST and CIFAR-10 dataset at the end of 30 and 100 epochs, respectively. First, we observe that the test error decreases steadily without any evidence of overfitting. Second, we see that the number of bits for each layer changes with the architecture. Third, we see that the test error is reduced by additional convolutional as well as dense layers. We observe good anytime performance as each of these experiments is run for only one hour, in comparison to 20 hours to get state-of-the-art results.

**Impact of Hyperparameters.** In this experiment, we show the impact of the regularization hyper-parameters in (4), $\lambda_1$ and $\lambda_2$. In this experiment, we train each CNN for 30 epochs only. Figure 4 shows the impact on performance and compression on the MNIST and CIFAR-10 data. In the figure, the compression ratio is defined as the ratio of the total number of bits used by LeNet32 ($= 32 \times 4$) to the total number of bits used by BitNet. In both the datasets, on one hand when $\lambda_2 = 0$ and $\lambda_1 = 1$, BitNet uses 32-bits that are evenly spaced between the range of parameter values. We see that the range preserving linear transform (1) leads to significantly better test error compared to LeNet32 that also uses 32 bits, which is non-linear and is not sensitive to the range. For MNIST in Figure 4 (left), BitNet with $\lambda_2 = 0, \lambda_1 = 1$, thus using 32 bits, achieves a test error of 11.18%, compared to the 19.95% error of LeNet32, and to the 11% error of BitNet with the best settings of $\lambda_1 = 10^{-3}, \lambda_2 = 10^{-6}$. The same observation holds true in the CIFAR-10 dataset. On the other hand, when $\lambda_1 = 0$ and $\lambda_2 = 1$, BitNet uses only 2-bits per layer, with a test error of 13.09% in MNIST, a small degradation in exchange for a 16x compression. This approach gives us some flexibility in limiting the bit-width of parameters, and gives an alternative way of arriving at the binary or ternary networks studied in previous work. For any fixed value of $\lambda_1$, increasing the value of $\lambda_2$ leads to fewer bits, more compression and a slight degradation in performance. For any fixed value of $\lambda_2$, increasing the value of $\lambda_1$ leads to more bits and lesser compression. There is much more variation in the compression ratio in comparison to the test error. In fact, most of the settings we experimented led to a similar test error but vastly different number of bits per layer. The best settings were found by a grid search such that both compression and accuracy were maximized. In MNIST and CIFAR-10, this was $\lambda_1 = 10^{-3}, \lambda_2 = 10^{-6}$.

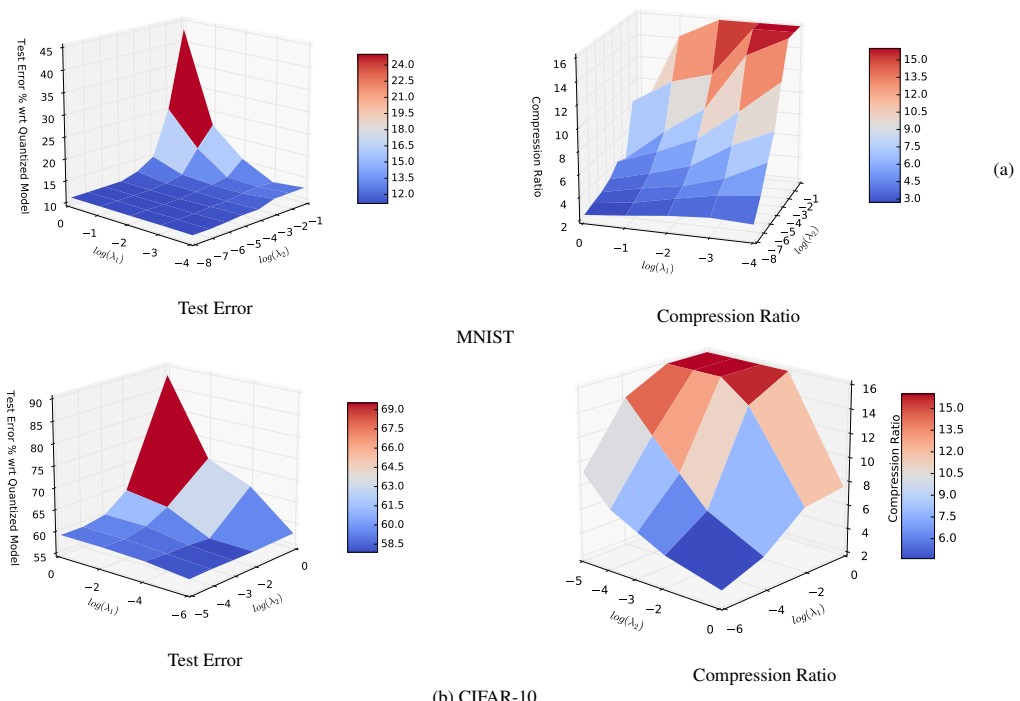

Figure 4: Sensitivity of the Test Error and Compression Ratio to the hyperparameters for BitNet for (a) MNIST and (b) CIFAR-10 datasets

## 6    CONCLUSION

The deployment of deep networks in real world applications is limited by their compute and memory requirements. In this paper, we have developed a flexible tool for training compact deep neural networks given an indirect specification of the total number of bits available on the target device. We presented a novel formulation that incorporates constraints in form of a regularization on the traditional classification loss function. Our key idea is to control the expressive power of the network by dynamically quantizing the range and set of values that the parameters can take. Our experiments showed faster learning measured in training and testing errors in comparison to an equivalent unregularized network. We also evaluated the robustness of our approach with increasing depth of the neural network and hyperparameters. Our experiments showed that our approach has an interesting indirect relationship to the global learning rate. BitNet can be interpreted as having a dynamic learning rate per parameter that depends on the number of bits. In that sense, bit regularization is related to dynamic learning rate schedulers such as AdaGrad (Duchi et al., 2011). One potential direction is to anneal the constraints to leverage fast initial learning combined with high precision fine tuning. Future work must further explore the theoretical underpinnings of bit regularization and evaluate BitNet on larger datasets and deeper models.

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
