# OpenReview forum: "Bit-Regularized Optimization of Neural Nets"
_ICLR.cc/2018/Conference — Reject_

### Official Review · AnonReviewer1 · 2017-11-26
**This paper focuses on how to train low-bit nets directly which is important. However, more experiments on large datasets are need to demonstrate the effectiveness of the proposed method.**

**Rating:** 4
**Confidence:** 5

**Review:**

This paper proposes a direct way to learn low-bit neural nets. The idea is introduced clearly and rather straightforward.

pros:
(1) The idea is introduced clearly and rather straightforward.
(2) The introduction and related work are well written.

cons:
The provided experiments are weak to demonstrate the effectiveness of the proposed method.
(1) only small networks on relatively small datasets are tested.
(2) the results on MNIST and CIFAR 10 are not good enough for practical deployment.

---

> ### Author Response · Authors · 2018-01-05
> **Rebuttal for AnonReviewer 1**
>
> 1- Only small networks on relatively small datasets are tested.
> >The results on VGG networks (larger networks) is being computed and will be included in the camera ready submission.
>
> 2-The results on MNIST and CIFAR-10 are not good enough...
> >We found that our low performance on MNIST was caused by using 4X4 pooling layers. We have changed this experiment to use a the standard neural net architecture as a baseline and results on MNIST are shown in (Figure 1a) and  CIFAR-10 (Figure 1b).

---

> ### Comment · AnonReviewer1 · 2018-01-10
> **Experimental results are still too weak.**
>
> 1. The state-of-the-art result on MNIST is clearly above 99.5% (top1 accuracy).  Actually by simply using LeNet, one can achieve top 1 accuracy higher than 99%. [1] However, in the paper the best result is less than 97%.
> 2. The state-of-the-art result on CIFAR is based on deep residual nets (eg.  wide residual nets can achieve less than 4% top 1 error rate).  And in the paper the best result is less than 90%.
> So, the current version is still too weak to demonstrate the effectiveness of the proposed method.
>
> [1] http://yann.lecun.com/exdb/mnist/
> [2] https://github.com/szagoruyko/wide-residual-networks

---

### Official Review · AnonReviewer3 · 2017-11-28
**The learning procedure seems to be wrong; experiments not comprehensive.**

**Rating:** 3
**Confidence:** 4

**Review:**

This paper proposes to optimize neural networks considering the three different terms: original loss function, quantization error and the sum of bits. While the idea makes sense, the paper is not well executed, and I cannot understanding how gradient descend is performed based on the description of Section 4.

1. After equation (5), I don't understand how the gradient of L(tilde_W) w.r.t. B(i) is computed. B(i) is discrete. The update rule seems to be clearly wrong.
2. The experimental section of this paper needs improvement.
   a. End-to-end trained quantized networks have been studied in various previous works including stochastic neuron (Bengio et al 2013), quantization + fine tuning (Wu et al 2016 Quantized Convolutional Neural Networks for Mobile Devices), Binary connect (Courbariaux et al 2016) etc. None of these works have been compared with.
   b. All the baseline methods use 8 bits per value. This choice is quite ad-hoc.
   c. Only MNIST and CIFAR10 dataset with Lenet32 are used in the experiment. I find the findings not conclusive based on these.
   d. No wall-time and real memory numbers are reported.

---

> ### Author Response · Authors · 2017-12-04
> **Re: The update rule seems to be clearly wrong.**
>
> Re: 1. After equation (5), I don't understand how the gradient of L(tilde_W) w.r.t. B(i) is computed. B(i) is discrete.
> This seems to be mistaken. Internally B(i) is a real number that is restricted to take integral values through this update rule using the sign function.
>
> It is easy to see how the gradient of L(tilde_W) is computed wrt B(i), ie. the gradient of q(W,B(i)) wrt B(i).  (n.b. q(.) is continuous and piecewise differentiable). To differentiate q wrt B(i) we need to differentiate tilde_W wrt B(i).
>
> For a fixed W, you can write tilde_W as a case expression with outputs \alpha+1\delta, \alpha+2\delta etc for different conditions for w \in W.
> dtilde_W/db is differentiated piecewise, and same as d\delta/db, \delta is a function of 1/(2**B(i)).

---

> ### Author Response · Authors · 2018-01-05
> **Rebuttal for AnonReviewer3**
>
> 1. After equation (5), I don't understand how the gradient of L(tilde_W) w.r.t. B(i) is computed. B(i) is discrete. The update rule seems to be clearly wrong.
> >The gradient update rule is correct, we added explanation that the number of bits is treated as a real number for the purpose of calculating gradients. The update rule ensures integrality. The next post discusses the details.
>
> 2-a. End-to-end trained quantized networks have been studied in various previous works... None of these works have been compared with.
> >We cite these papers in our related work section. We would like the reviewer to note that the networks, in the papers referred to, are trained with a fixed (hand coded) number of bits. The research question we are answering is: What is the optimal number of bits? The research question these references answer which is: What is the performance given a certain fixed number of bits.
>
> 2-b. All the baseline methods use 8 bits per value. This choice is quite ad-hoc.
> >We have added baseline experiments using 4 and 16 bits (Figure 1 a&b). We have to fix the number of bits for these algorithms.
>
> 2-c. Only MNIST and CIFAR10 dataset with Lenet32 are used in the experiment. I find the findings not conclusive based on these.
> >Most other approaches on low precision training such as (BinaryConnect by Courbariaux et al NIPS 2015) and (Soft weight sharing for NN Compression by Ullrich et. al. 2017) only compare on these simple datasets.
>
> 2-d. No wall-time and real memory numbers are reported.
> >We are unclear whether this is regarding training or inference time. There is about 4X savings. Re inference time: a meaningful comparison requires hardware support for low precision operations. This is currently unavailable for arbitrary precision.

---

### Official Review · AnonReviewer2 · 2017-11-30
**Interesting idea; insufficient analysis of training methodology and concerning empirical work**

**Rating:** 4
**Confidence:** 4

**Review:**

The paper proposes a technique for training quantized neural networks, where the precision (number of bits) varies per layer and is learned in an end-to-end fashion. The idea is to add two terms to the loss, one representing quantization error, and the other representing the number of discrete values the quantization can support (or alternatively the number of bits used). Updates are made to the parameter representing the # of bits via the sign of its gradient. Experiments are conducted using a LeNet-inspired architecture on MNIST and CIFAR10.

Overall, the idea is interesting, as providing an end-to-end trainable technique for distributing the precision across layers of a network would indeed be quite useful. I have a few concerns: First, I find the discussion around the training methodology insufficient. Inherently, the objective is discontinuous since # of bits is a discrete parameter. This is worked around by updating the parameter using the sign of its gradient. This is assuming the local linear approximation given by the derivative is accurate enough one integer away; this may or may not be true, but it's not clear and there is little discussion of whether this is reasonable to assume.

It's also difficult for me to understand how this interacts with the other terms in the objective (quantization error and loss). We'd like the number of bits parameter to trade off between accuracy (at least in terms of quantization error, and ideally overall loss as well) and precision. But it's not at all clear that the gradient of either the loss or the quantization error w.r.t. the number of bits will in general suggest increasing the number of bit (thus requiring the bit regularization term). This will clearly not be the case when the continuous weights coincide with the quantized values for the current bit setting. More generally, the direction of the gradient will be highly dependent on the specific setting of the current weights. It's unclear to me how effectively accuracy and precision are balanced by this training strategy, and there isn't any discussion of this point either.

I would be less concerned about the above points if I found the experiments compelling. Unfortunately, although I am quite sympathetic to the argument that state of the art results or architectures aren't necessary for a paper of this kind, the results on MNIST and CIFAR10 are so poor that they give me some concern about how the training was performed and whether the results are meaningful. Performance on MNIST in the 7-11% test error range is comparable to a simple linear logistic regression model; for a CNN that is extremely bad. Similarly, 40% error on CIFAR10 is worse than what some very simple fully connected models can achieve.

Overall, while I like the and think the goal is good, I think the motivation and discussion for the training methodology is insufficient, and the empirical work is concerning. I can't recommend acceptance.

---

> ### Author Response · Authors · 2018-01-05
> **Rebuttal for AnonReviewer2**
>
> 1- It's also difficult for me to understand how this interacts with the other terms in the objective (quantization error and loss)... it's not at all clear that the gradient of either the loss or the quantization error w.r.t. the number of bits will in general suggest increasing the number of bit.
> >We have clarified this in the current revision. It is reasonable to assume that the classification accuracy is similar for similar values of the parameters. The error due to the local linear approximation drops at a rate of 1/2^B. In the worst case, we use a fine grained approximation using 32 bits. This is clearly not needed as we show in the experiments, that a 5-6 bit local linear approximation gives good accuracy.
>
> 2- It's unclear to me how effectively accuracy and precision are balanced by this training strategy.
> >We have clarified this in the current revision. As we show in our experiments, the accuracy and precision trade-off varies with different values for \lambda_1 and \lambda_2.
>
> 3-The results on MNIST and CIFAR10 are so poor that they give me some concern about how the training was performed and whether the results are meaningful.
> >We believe the reviewer is referring to Table 1. These are after 30 epochs only for MNIST. Our final error rate was 3% on MNIST. Please note that we are using a small learning rate (1e-3) in order to show the big impact of bit regularization. In Figure 3(a)(right panel), we showed that increasing the learning rate does indeed improve the performance to about a 2% error rate.
>
> >In the initial submission, we showed LeNet32 and BitNet with about 4% test error rate. We found that our low performance on MNIST was also caused by using 4X4 pooling layers. We have changed this experiment to use a 2X2 pooling and now show error rates of 2-3% on MNIST (Figure 1a) and 27-29% on CIFAR-10 (Figure 1b).

---

### Public Comment · (anonymous) · 2017-11-18
**Source code & implementation details**

Hello, I am working on reproducing your work for the ICLR 2018 Reproducibility Challenge. I was wondering if/when you plan to open source the code used to perform your experiments.
I would also appreciate if you could provide more details on the model architectures used by you.

Thanks and best regards

---

> ### Author Response · Authors · 2017-11-23
> **Re: implementation**
>
> Thank you for your comment. We have not open sourced our implementation yet. We will get back to you after the review period.

---

### Author Response · Authors · 2018-01-05
**General Rebuttal**

We thank the reviewers for their feedback.  AnonReviewer1 clearly understood the paper saying "The idea is introduced clearly and rather straightforward." AnonReviewer2 also seemed to fully understand the paper and its contributions saying "the idea is interesting, as providing an end-to-end trainable technique for distributing the precision across layers of a network would indeed be quite useful." AnonReviewer3 missed a couple of key points in our approach, which made the reviewer think the formulation is incorrect.

This revision contains the following modifications and we reference these modifications accordingly in each of the reviewer's rebuttal:
1- All three reviewers raised the following concerns about the experiments:
The performance on MNIST and CIFAR doesn't match the state-of-the-art.
>We found that our performance can be improved by using 2X2 pooling instead of 4X4 pooling. Please note that our experiments use a small learning rate (1e-3) because it better distinguishes BitNet from LeNet. We showed in Figure 3(a)(right panel) that increasing the learning rate does lead to better accuracy. We have changed this experiment and now show error rates of 2-3% on MNIST (Figure 1a) and 27-29% on CIFAR-10 (Figure 1b).

2- AnonReviewer3 mentioned that there was no variation in base line experiments with regards the number of bits ( only focusing on 8bits)
>We added baseline experiments using 4 and 6 bits (Figure 1a and 1b).

3- Overall clarity of the text and formulation
>We added overall text and formulation clarification
1- Added explanation that the number of bits is treated as a real number for the purpose of calculating gradients.
2- Added the closed form of the gradients of the quantization error wrt the number of bits and the weights.
3- Expanded gradients in Eq (5) to show gradients wrt the terms of the proposed loss function.
4- We added some clarifying text towards the end of Section 4.

---

### Decision · Program_Chairs · 2018-01-29
**ICLR 2018 Conference Acceptance Decision**

**Decision:**

Reject

**Comment:**

Pros:
+ The idea of end-to-end training that simultaneously learns the weights and appropriate precision for those weights is very appealing.

Cons:
- Experimental results are far from the state-of-the-art, which makes the empirical evaluation unconvincing.
- More justification is needed for the update of the number of bits using the sign of the gradient.